# Super-Resolution Microscopy of Chromatin

**DOI:** 10.3390/genes10070493

**Published:** 2019-06-28

**Authors:** Udo J. Birk

**Affiliations:** 1University of Applied Sciences HTW Chur, Pulvermühlestrasse 57, 7004 Chur, Switzerland; udo.birk@htwchur.ch or birku@uni-mainz.de; Tel.: +41-(0)81-286-3797; 2Institut für Physik, Universität Mainz, 55128 Mainz, Germany

**Keywords:** chromatin, super-resolution microscopy, nuclear architecture, single molecule localization microscopy, DNA, fluorescence, DNA labeling

## Abstract

Since the advent of super-resolution microscopy, countless approaches and studies have been published contributing significantly to our understanding of cellular processes. With the aid of chromatin-specific fluorescence labeling techniques, we are gaining increasing insight into gene regulation and chromatin organization. Combined with super-resolution imaging and data analysis, these labeling techniques enable direct assessment not only of chromatin interactions but also of the function of specific chromatin conformational states.

## 1. Introduction

Undoubtedly, the cell nucleus is a highly organized organelle. A myriad of molecular processes help to maintain its integrity and function. An individual process requires the interplay of several components, which quite likely need to be positioned in close proximity to each other. For instance, if transport processes across the nuclear envelope are involved, these processes eventually require components to be located in close vicinity to respective nuclear transport portals such as e.g., nuclear vesicles [1] or nuclear pores [2,3]. Additionally, protein folding, and hence the protein shape or form, is known to affect binding and transport, as clearly seen in the movement of myosin motors along actin filaments [4]. Similarly, the formation of a biomolecular machine by combining several proteins into a larger form or complex offers advantages, such as e.g., multiple DNA transcription sites as obtained by combining multiple polymerase II proteins into a larger complex [5]. As illustrated by these examples, positioning, form and function of the participants involved in the numerous processes are often mutually beneficial. While on the length scale of the individual molecule this is generally accepted, the implications of nuclear architecture on nuclear functions and vice versa on a larger scale are an active research topic, giving rise to scientific debates. To mention only a few of the questions at hand: to what extent does the structuring of chromatin in topological regions [6,7,8] (e.g., chromatin domains) contribute to nuclear functioning? On what length scales does this structuring occur [9]? How is it dynamically organized [10,11] e.g., during the cell cycle or during DNA replication or transcription? How are the dynamical processes themselves responsible for the possibly induced structural changes of chromatin [9]? Researchers have coined the term “4D nucleome” [12,13] to express the whole range of dynamic, biophysical and biochemical properties of chromatin. Under the term “4D nucleome”, we may include the structure–function interactions of chromatin domains, regulatory sequences and genes with each other in *cis* and *trans*, as well as all the countless genetic and epigenetic interactions with functional machineries and nuclear bodies. 

We might expect that we could better understand processes on the macroscale, such as the development and treatment of disease, if we can link these processes to differences in nuclear phenotypes between normal and pathologically altered cell types. In addition to the study of aberrant nuclear phenotypes, also more generally the study of basic nuclear functions could benefit from linking function to the nuclear topography. Firstly, transcription regulatory elements play a crucial role in the regulation of genetic activity. Secondly, regulation of genetic activity can be also accomplished by shielding regulatory elements from attachment of the respective activator or inhibitor, as it is realized e.g., in highly condensed i.e., compacted chromatin [14,15], which is inaccessible to large molecules. Thirdly, the binding of proteins to several binding sites can lead to the formation of a large-scale chromatin network [16]. For these reasons, it appears natural to investigate the dynamic structure of chromatin on a multitude of length and time scales, in order to gain further insight into the relationship between form, positioning and function of these highly convoluted multi-molecular complexes [17,18].

As a tool for our investigation of chromatin, we would like a method that fulfills a number of requirements: naturally, the investigation should not alter the sample, so that it remains in its original shape. Additionally, the cells need to be kept in their “natural” state, in order to maintain proper functioning. For living cells, a good control is that the cell is continuing to divide after our investigation. However, we would like the investigation to capture the dynamics of rapid processes, so we need to be very fast. The ideal measurement candidate will of course allow error-free quantitative analysis, and not introduce additional ambiguities when it comes to data evaluation. It needs to provide the highest possible imaging resolution. Biomedical image-based analyses traditionally achieve good contrast using contrast agents. For microscopy, the agent of choice is often fluorescence, allowing the separation of individual types of components (e.g., specific proteins) based on the color emitted by the attached fluorophores. In terms of 3D extent of the samples, our chromatin-investigation method should not pose too severe constraints. If the object were large, we could possibly make use of a suitable scanning technique that allows us to investigate the sample bit by bit; if the objects were thick, we would need a method providing sectioning capability (suppression of signals from out-of-focus areas) as well as 3D enhanced resolution. As we expect our method of choice not to be foolproof, it will certainly be beneficial if we can combine it with other imaging modalities.

When we speak about non-invasive tools, a whole range of microscopy methods spring to mind, each of them with its own strengths and weaknesses. Light microscopy in particular, appears to be very appealing, for its ease of use, its high specificity when combined with fluorescence labels, its potentially low impact on the sample, its live cell imaging capability, and its good penetration and transmission through cell samples. As we shall see in the following, however, the fulfillment of our list of requirements is not easily met. A number of future developments need to be realized before we can move along the path of quantitative assessment of chromatin structure at length scales ranging from a few micrometers down to the nanometer level [19].

Nonetheless, a number of imaging methods with an effective resolution below the Abbe limit as outlined below have been devised; these techniques have been termed ‘super-resolution’ microscopy [20] to highlight their capability to provide super-resolution. In the following, I will concentrate on the discussion of super-resolution in the context of far-field fluorescence microscopy applied to image chromatin. In the next section, we will discuss what challenges we face when performing microscopic experiments, and how these challenges may be overcome. In Section 3, individual super-resolution microscopy techniques with example applications to image chromatin will be presented. Namely, the techniques of stimulated emission depletion (STED) microscopy, structured illumination microscopy (SIM), and some of the single molecule localization microscopy (SMLM) techniques will be discussed.

## 2. Challenges in Microscopy of Chromatin

In the study of chromatin organization in intact cells, advanced cell-imaging techniques have proven most helpful. Cryo-electron tomography [21,22] as well as super-resolution light microscopy methods [23] have immensely contributed to our understanding of the spatial and dynamic organization of chromatin. Results obtained by applying these highly effective imaging methods indicate the existence of a range of chromatin compaction states with a degree of heterogeneity [21,24,25,26,27,28,29]. Furthermore, procedures such as fluorescence correlation spectroscopy (FCS) [30,31] and fluorescence anisotropy [32], which make use of fluorescently tagged histones [3,33], allow the dynamics of chromatin to be studied. Since the advent of super-resolution microscopy, numerous approaches to study the spatial and temporal chromatin organization, using the various super-resolution imaging approaches have been published. In the following, a discussion of major obstacles to be overcome in these applications is presented.

### 2.1. Limited Imaging Resolution

In 1873, Ernst Abbe concluded from his studies, that an object under observation has an impact on the light passing through the object, namely that the light is affected by so-called “diffraction”, and that this determines whether an object structure can be transmitted into the microscopic image or not. The term diffraction is used to illustrate that light bouncing off small structures is travelling in distinct directions characteristic of the size of the structures: The smaller the structures, the larger the angle between the incident light and the outgoing light. In microscopy, this is quite unfortunate, as the microscope objective lens does not capture light under very high angles, such that information about the very small structures can never be passed through the objective lens, unless very particular methods of microscopy are being used. 

The findings of Ernst Abbe clearly marked that the smallest structures, which may be resolved by imaging through an objective lens, are in the order of 0.2 µm. For more than a century, this was the generally accepted limit for working with the microscope. Beyond this optical resolution limit, finer details such as the composition of molecular complexes cannot be resolved. The parameters responsible for the different resolution values attainable are the wavelength of the light used for imaging, and the numerical aperture (indicative of the acceptance angle) of the microscope objective lens. Interestingly, none of the super-resolution microscopy methods have proven Abbe’s considerations on the fundamental limits of optical resolution to be wrong. However, the combination of fluorescence excitation and emission, together with the Abbe resolution limit, can be used to emboss additional information in the detected fluorescence signal, thereby allowing us to discriminate between unresolvable neighboring object structures. For example, in STED and SMLM, the neighboring molecules under suitable conditions can be made to emit their fluorescence signal at different times. Therefore, these detected signals can be discriminated and individually assigned to distinct emitters, even if their mutual distance is below the resolution limit.

With the highly sensitive detectors invented starting in the 1990s, it became possible to routinely detect the fluorescence signal from tiny populations of individual fluorophores, and even single fluorophores. As can be expected, the signal-to-noise ratio (SNR) is the fundamental parameter not only for the signal detection, but for the subsequent analysis of the signal. As in classic optical microscopy, a fundamental factor influencing the SNR is the optical resolution limit, the other two parameters are the absolute amount of light attributing to the fluorescence signal and the amount of light (or noise) present in areas where we would expect no fluorescence emission. These three factors are then responsible for the final detection accuracy in super-resolution microscopy.

### 2.2. Limited Contrast

The huge success that fluorescence microscopy has seen in biomedical studies since it has been introduced is mostly based on the improved contrast which is achieved by joining fluorescence labels solely to the structure of interest, and subsequently detecting only structures which were effectively labeled on a virtually non-existing (i.e., zero) background signal. In order to facilitate this, the light emitted by fluorophores has a different wavelength than the light used for excitation. Using highly selective filters, the excitation light is blocked from reaching the detector. This allows us to detect the emitted fluorescence light, which is typically orders of magnitude weaker. A single fluorophore may emit fluorescence light several millions of times, however, typically, only a tiny fraction of the emitted light is detected and converted to a useful signal. This severely limits the signal strength, and for lack of suitable detectors has for many years hindered researchers from detecting individual fluorophores by their fluorescence emission.

Over the last few years, a few other contrasting strategies have been inferred, based on the advanced understanding of the photophysics of fluorescence in biological samples. Several parameters have an impact on the fluorescence emissivity. This allows us to control the fluorescence process inside the cell sample, which is one of the requirements used in many super-resolution microscopy approaches. Parameters that characterize fluorescence emissivity can be many, including the following list. The absorption cross-section is responsible for how easily the fluorophore gets excited. Only after excitation can the fluorescence light be emitted. The quantum yield tells us how likely the fluorophore is to emit light once it is excited. Other processes compete with the fluorescence emission, depending e.g., on collision with other molecules, on transport of electrons, and on dissociation of the fluorophore. The lifetime of a fluorophore is limited. As the molecule experiences a number of excitation and subsequent fluorescence emission processes, there is an increasing likelihood for the molecule to lose its capability to emit fluorescence, e.g., due to chemical reactions. Thereby, this fluorophore is expulsed from the population of accessible, i.e., emanating, fluorophores within the sample, resulting in a dimmer appearance of the stained cells. Often, this process is induced by light, hence the process is termed “photobleaching”. Bleaching is notably (positively or adversely) influenced by the chemical environment, in which the fluorophores are imaged. A major cause for loss of fluorescence is often attributed to the fact that the re-emission of the fluorescence light takes place only after a period of time. In the time between excitation and fluorescence emission, the fluorophore has been transferred to a higher energy state; energy which can be used to destroy the fluorescent molecule.

#### 2.2.1. High Sensitivity Based on Bright and Stable Emitters

The basic requirements on the fluorophores are that they must be photostable and bright. The product of the efficiency, with which the fluorophore converts absorbed light to fluorescence light (“quantum yield”), and the efficiency, with which the fluorophore absorbs light (“extinction coefficient”), is usually defined as the “brightness” of the fluorophore. Consequently, the brightness is essentially a measure for how much light is emitted by the fluorophore. 

In super-resolution microscopy, the brightness of a fluorophore has an influence on several levels during signal detection and evaluation: brighter fluorophores are simpler to distinguish, and largely boost the signal-to-noise ratio. This has the potential side effect of facilitating experiments at decreased acquisition time, which is particularly imperative for the study of highly dynamic processes in live cells. The ideal fluorophore emits fluorescence light whenever it was excited, aiming at a quantum yield equal to one.

Nowadays, a huge range of chemistries for labeling techniques is accessible, including labeling of proteins, peptides, antibodies, manufactured oligonucleotides, ligands, and other biomolecules. Commonly utilized strategies comprise immunochemistry (i.e., labeling with the help of antibodies), genetic transfection with fluorescent proteins, different methodologies for labeling DNA (e.g., fluorescence in situ hybridization (FISH)), and direct labeling with ‘click’-chemistry (e.g., DNA base analogue such as 5-ethynyl-2'-deoxyuridine, EdU [34,35]). In antibody-labeling, a common issue has been the size of the antibody itself, usually inducing a distance between the fluorophore and the labeling target site. With the introduction of single domain antigen-binding fragments, also known as ‘nanobodies’, this is much less of an issue. With molecular sizes of typically around 15 kDa, nanobodies are considered perfect candidates for numerous super-resolution imaging applications.

#### 2.2.2. High Specificity Based on Stable Binding of the Emitters to Their Target

The fluorescent tags utilized in super-resolution microscopy need to be coupled to a specific target site. In a perfect world, this coupling should be not only highly selective and efficient, but also stable or at least transiently stable, so that a sufficient amount of light can be collected while the fluorophores are residing at their correct position. Moreover, the link distance between the fluorophores and the target structure ought to be as small as possible; a few improvements worth mentioning in this direction are ‘click-chemistry’ and ‘nanobodies’ [34].

**Figure 1 genes-10-00493-f001:**
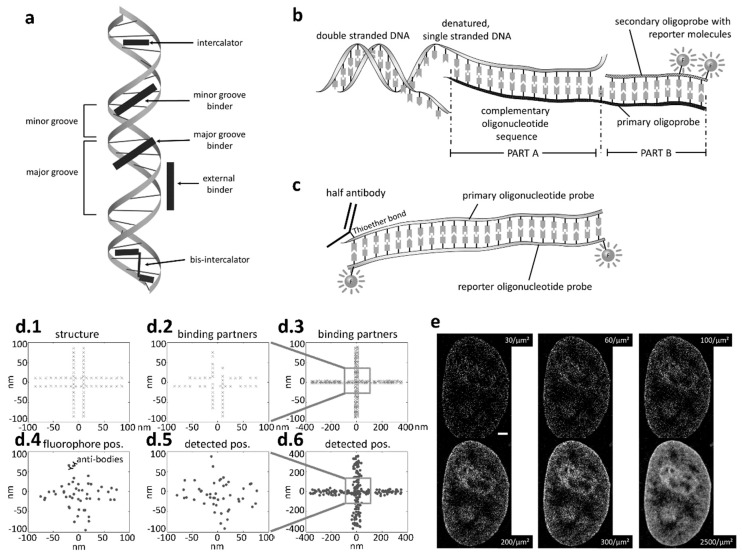
Imaging DNA. (**a**) Apart from indirect labeling via tagged proteins, DNA can be labeled directly using a number of labeling strategies with DNA specific dyes binding e.g., to the minor or major grove, or in-between the two DNA strands (intercalating dye). (**b**) Oligonucleotide sequences offer the possibility to attach a removable fluorescently labeled oligoprobe to single stranded DNA, e.g., after a denaturation step [36]. (**c**) The fluorophore is often attached to the end of the oligonucleotide. The technique can not only be used to label DNA, but also be combined with immunolabeling e.g., with functional antibody fractions. (**d**) When labeling a structure such as the DNA (d.1), often some binding partners are left unlabeled (d.2), and the linker (e.g., antibody) between the fluorophore and the target site (d.4), as well as the finite imaging precision (d.5), result in a deteriorated reconstruction (d.6). (**e**) Single molecule localization microscopy (SMLM) of nuclear DNA labeled with Hoechst [37]. Scale bar: 2 µm. The effect of limited labeling efficiency when imaging chromatin structure is clearly visible, with better image quality obtained when the number of detected fluorophore signals is increased from 30 to 2500 per µm².Due to abundancy, structural resolution could possibly be much higher when DNA specific dyes are used as opposed to labeling DNA bound proteins [38]. © 2017 WILEY-VCH Verlag GmbH & Co. KGaA. Reprinted from [23] with permission from Wiley.

Various approaches for sequence specific imaging of DNA [23] are shown in Figure 1. Typically such approaches are called ‘DNA tags’ or ‘DNA labels’, and are briefly discussed below. A better insight into the topic might be obtained through recent review articles on labeling and imaging chromatin in live cells [39,40,41]. Specific labeling includes fluorescence labeling with aptamer- or oligo-probes. Additionally, there is also direct labeling using CRISPR (clustered regularly interspaced short palindromic repeat) or TALE-N (artificial nuclease based on transcription activator-like effectors) techniques [42,43]. Repetitive sequences are more often labeled using TALEs while CRIPSR/Cas is more useful in labeling non-repetitive sequences. Both are compatible with super-resolution STED and SMLM. Other methods include e.g., the SunTag method or the more traditional methods for staining DNA with common DNA-specific fluorophores such as YOYO, Hoechst, DAPI, and many more. The concept of fluorescence in situ hybridization (FISH) with oligo probes has been further developed to facilitate labeling with removable oligo-nucleotide tags [36]. 

### 2.3. Limited Temporal Resolution

#### 2.3.1. Fast and Sensitive Detection

Modern array detectors such as those used in scientific cameras provide fast read-out of large amounts of image data at very high speed with limited noise added due to the cooled detector. Still, frame rates are seldom above a few 100 frames per second because of the time which is needed to collect sufficient fluorescence light from a small sample volume (ideally a single fluorophore). Critical factors limiting the frame rate are e.g., the illumination intensity (responsible for excitation of the fluorophores) and the time that the fluorophore requires before it re-emits the fluorescence light.

Fast and sensitive detection schemes do not necessarily rely on imaging approaches. Sometimes, fluorescence correlation spectroscopy (FCS) is better suited to study the interaction between partner molecules. Additionally, Förster resonance engergy transfer (FRET) is a suitable means of studying the distances between interaction partners. 

#### 2.3.2. Fast Reconstruction Software

A large variety of evaluation algorithms is available for the different super-resolution microscopy techniques. The ideal data evaluation algorithm provides error-free rendering and quantitative analysis of the acquired microscopy data in real-time, to support the live cell imaging approaches of fast dynamic processes. Modern computer hardware has greatly facilitated the rapid reconstruction of large amounts of data; with the help of parallelization such as implemented on modern graphical processing units (GPUs), the rendering of the data becomes available with little temporal delay.

As a result of the large number of available data evaluation software and the different types of super-resolution microscopy methods, the quantitative numbers extracted by supposedly similar experiments may vary significantly, as indicated e.g., in the various discussions related to the attained effective imaging resolution [44,45,46]. Other quantitative parameters are sometimes extracted with similar ambiguities, which makes it difficult to conclude which software should be used in a particular experimental setting, not only for the beginner but also for the application expert. A good software comes with intrinsic correction and parameter estimation algorithms in order to assist the users in their choices. Such problems have been successfully solved by the communities of confocal laser scanning microscopy [47] and light sheet fluorescence microscopy [48], where the community develops layout and implementation of open source software, and publicly discusses the essential requirements for the implementation of best practice analytics. While software and algorithms for super-resolution microscopy are regularly reviewed [49,50], open discussion and best practices are still under development.

### 2.4. Structural Integrity

Ideally, super-resolution imaging is applied to live cells. Sometimes, due to the requirements of the labeling strategy, imaging of fixed samples cannot be circumvented. In any case, in order to visualize chromatin ultrastructure, it is necessary to carefully consider the sample preparation and imaging protocols with respect to the one goal: structural integrity. Only if the chromatin structure of interest is preserved throughout the experiment, the microscopy data will further our understanding of the underlying biological processes. Many of the sample properties are affected by the sample preparation and imaging protocol. Apart from the aforementioned considerations on the structural integrity, the embedding medium in particular has to help with three important tasks: 1) maximize the fluorescence signal, 2) minimize the fluorescence background, and 3) minimize optical aberrations.

#### 2.4.1. Labeling Artifacts Such as Sterical and/or Functional Hindrance

A variety of labeling strategies to equip proteins with a fluorescent tag is available. However, the labeled proteins might not have the same behavior as their unlabeled counterparts. Typically, control experiments are designed to verify that there are no noticeable differences between the population of labeled cells as compared to unlabeled cells. Nonetheless, cells as complex biological systems could have multiple pathways to achieve the same goal, thereby shielding the effect of the fluorescence label on the protein function. Apart from that, proteins carrying a fluorescent label might still be functional, but their size could have been modified in such a way, that their dynamic behavior (as e.g., defined by the diffusion or active transport) is severely altered. Finally, immunochemistry, i.e., labeling with antibodies, employs one of the largest possible label carriers, which on top can form cross-links with other antibodies, such that although much more material is physically added to a structure of interest, the overall size and shape of the structure is decreased [51]. 

#### 2.4.2. Harsh Chemical Treatments

Chemical fixation and permeabilization in fixed cell experiments can give rise to artifacts. A debate as to the introduction of artifacts when using either formaldehyde fixation or methanol acetic acid fixation has been going on for years. The latter is known to flatten the nucleus, potentially resulting in severe distortion of the 3D structure, although structural changes on the nanometer scale might not be so dramatic after all. Sometimes, the fixation protocols introduce additional reagents to improve preservation of the cell structure, albeit at the cost of adversely affecting the fluorescence signal. The harsh permeabilization, rehydration and dehydration treatment with alcohol, heat or pH-based denaturation in FISH and oligo-labeling, and RNase treatment, can lead to a significant damage to the 3D chromatin structure. Furthermore, the cross-linking process in the fixation step might be slow compared to the protein association with or dissociation from the DNA. 

#### 2.4.3. Mechanical and Chemical Effects of Cell Separation: Cells Outside Their Natural Environment

Isolated cells are ideal samples for light microscopy because they are transparent and thin. However, a cell in its isolated form is likely to behave differently from cells in tissue. For this reason, large efforts are put into imaging of cell aggregates, such as e.g., spheroids. However, as these aggregations are no longer transparent, many of the super-resolution microscopy methods cannot easily be employed.

#### 2.4.4. Swelling and Clearing

As an alternative to structure preservation techniques, a number of somewhat counterintuitive approaches have been developed, in which the biological samples are first subjected to a systematic swelling procedure. The reason is as simple as it is effective: the limited optical resolution of light microscopy does not allow investigation of small structures. If the structures were much larger, the limited resolution could still be sufficient to investigate the problem at hand. One of these techniques has been termed ‘expansion microscopy’ [52]. Clearing helps to reduce residual scattering [23].

## 3. Techniques

Minimally invasive sample preparation methods are required to preserve chromatin structure [53], such methods combined with super-resolution fluorescence microscopy allow the investigation of chromatin organization in live cells. Often, super-resolution microscopy methods are divided into three major categories: structured illumination microscopy (SIM, [54,55]), single-molecule localization microscopy (SMLM, [56,57]), and stimulated emission-depletion microscopy (STED, [58]). Potentially, SIM is better suited for multi-color live-cell experiments, while SMLM and STED allow imaging at higher spatial resolution.

### 3.1. Structured Illumination Microscopy (SIM)

Instead of homogeneously illuminating the sample as in classical widefield microscopy, structure illumination microscopy approaches employ a highly structured illumination pattern across the field of view. The pattern is shifted by small amounts resulting in a fluctuation of the fluorescence signal. The fluctuation contains additional information about the investigated objects, resulting in a two-fold increase in optical resolution, yielding a resolution of approximately 100 nm in-plane and 250 nm along the direction of viewing.

SIM has also become an effective tool to evaluate the spatial distribution of chromatin organization on the 100 nm length scale and for studying DNA-associated protein complexes related to DNA loop stability and DNA repair [59,60,61]. Best images are usually obtained when combining SIM with smart illumination schemes (see Section 3.4 Smart illumination schemes).

### 3.2. Stimulated Emission Depletion Microscopy (STED)

In a few applications, STED has been effectively utilized to image chromatin structures. In one case, STED was used to image DNA stained with the far-red DNA dye. As DNA label, SiR-Hoechst was used, which is a bisbenzimide-SiR conjugate to the bisbenzimide center of Hoechst 33342 [62]. In these applications, live HeLa cells were imaged in a commercial STED microscope at an excitation wavelength of 775 nm. In this live-cell application example, chromatin structures were imaged using STED nanoscopy with resolution clearly below 100 nm. Apart from this study, very little research has been published on the application of STED to image chromatin [63,64].

### 3.3. Single Molecule Localization Microscopy (SMLM)

Most applications targeting chromatin have been seen using SMLM as the super-resolution method of choice. Many labeling strategies have been optimized to be compatible with SMLM. Also, smart illumination schemes are available in order to avoid out-of-focus blur contributions and unnecessary bleaching of the fluorescent probes. Particular illumination and/or detection schemes can also be employed to provide the 3D position information on the individual fluorophores. Under good conditions, 10–20 nm resolution can be achieved in 3D SMLM. Reconstruction of an SMLM image requires a number of assumptions on the underlying fluorophore emission profiles. Typically, the emitters need to be isolated in space and time, i.e., only one (or few) emitters can be allowed to simultaneously emit light, while they can still be detected as individual emitters.

Discrete “clutches” of chromatin were found using SMLM in combination with labeled H2B histone proteins and DNA dyes [25,38]. Compact nucleosome spaces were shown to move through the cell nucleus [65], and polymerase binding has been shown to induce a large scale structure in the cell nuclues [16]. The co-localization of CCCTC-binding factor (CTCF) and cohesion proteins was shown using SMLM [60]. Several post-translational histone modifications have been visualized using SMLM in order to assess chromatin structures at different epigenetic states [28,66]. SMLM has moreover been broadly utilized in combination with modern DNA labeling strategies to elucidate the chromatin structure in situ. For instance, stress induced changes on large-scale nuclear architecture could be visualized using SMLM [27].

### 3.4. Smart Illumination Schemes

As outlined above, the highest precision in the analysis of super-resolution microscopy images is obtained when there is virtually no background. SIM and SMLM however, rely on wide-field detection, which allows contributions from out-of-focus volumes to be detected as blurred out background signal. To circumvent this, researchers have employed particular illumination schemes in order to facilitate fluorescence excitation inside the focal plane only. This technique is known from light-sheet fluorescence microscopy (LSFM) and related approaches [67,68,69], and can readily be combined with illumination grid patterns as required for SIM [70]. Apart from reducing the out-of-focus blur, the cells are much less exposed to illumination light, therefore the technique is muss less invasive. As an alternative to light sheet illumination, total internal reflection fluorescence (TIRF) excitation can be used to restrict fluorescence excitation to an extremely thin layer typically close to the interface between cover glass and sample. As a third approach, a kind of combination of TIRF and light-sheet illumination is sometimes employed: The object is illuminated through the objective lens with a light sheet with a high inclination with respect to the optical axis [71,72]. 

When used in combination with SMLM or SIM, light-sheet fluorescence microscopy not only suppresses out-of-focus light, but also results in larger imaging depths [73,74]. The section thickness from which fluorescence is detected can be dramatically reduced, and allows imaging of samples more than 100 µm thick [73]. In addition, axially structured illumination e.g., using a mirror [75] or model based approaches [76] can be employed to enhance measurement of position and extension of replication and transcription complexes and gene domains [3,5,51].

### 3.5. Statistical Analyses

Statistical analyses of the large number of images acquired in SMLM or similar acquisition protocols allow us to extract additional information, e.g., to better isolate the fluorescence emitters in a cluster or to increase the optical resolution in general [77,78,79]. Such methods make use of the fact that the fluorophore can only occupy distinct states (excited or disexcited), and that the transition of the fluorophore always involves both states, i.e., the fluorophore cannot go from excited to excited state without in-between being in a disexcited state.

### 3.6. Deep Learning

State-of-the-art deep learning methods employ e.g., generative adversarial networks (GAN) for enhanced reconstructions of SMLM data [80], which are less affected by dense emitter artifacts. In principle, trained artificial neural networks can provide reconstructions from conventional confocal laser-scanning microscopy or total internal reflection fluorescence (TIRF) microscopy data similar to those obtained by high-resolution STED or TIRF-SIM [81].

### 3.7. Single-Molecule Dynamics

The mobility of reporter molecules provides another means for studying structure. As an example, the volume accessible to a fluorophore can be imaged by tracking the fluorophore over time [82]. In FRET, the binding behavior of two partners is studied, which can be used for real-time detection of chromatin conformational dynamics (Figure 2) [83].

### 3.8. Correlative Microscopy

Electron microscopy (EM) has a much higher resolution, but cannot easily achieve the same contrast with the same number of different labels available in fluorescent microscopy. In many respects, the data obtained when using both imaging methods in combination can be superior [84]. There are not many labels specific for visualizing DNA in EM, however, a DNA labeling method termed ChromEMT has recently been published [21]. Sometimes also different types of super-resolution light microscopy are combined to provide high-quality images with intrinsic control [85].

### 3.9. Specific Considerations Needed for Imaging Chromatin

The fluorophores used in super-resolution microscopy are required to undergo a larger number of transitions between excited and ground state, in order to emit a signal that is sufficiently large to be detected and analyzed. In consequence, rapid bleaching of the fluorophore attached to the chromatin occurs. For this reason, as well as in order to increase contrast, many of the genome imaging approaches require dye molecules to bind dynamically to their chromatin target site [86,87,88]. Accordingly such methods work with molecules that exhibit suppressed fluorescence output when the dye molecule is not bound, either by smearing out the fluorescence signal during imaging as in traditional points accumulation for imaging nanoscale topography [89,90], or by quenching of the unbound DNA dye [38].

High precaution is indicated when interpreting the results from super-resolution microscopy images of chromatin. As the chromatin content in intact cell is high, the signals from fluorophores attached to neighboring structures, e.g., in densely packed volumes, are either impossible to generate, as the fluorophore accessibility during labeling could be affected by sterical hindrance, or impossible to separate up to the point where they need to be treated as a combined electronic entity. This fact is used in FRET (c.f. Figure 2), but might occur in forms of fluorescence microscopy using high concentrations of a single type of fluorophore as well. Similarly, absorption is an issue in high concentration fluorophore imaging. In general, chromatin density inside the nucleus varies to a large degree [27], and folding has a high impact on labeling, accessibility and detection of the fluorophores [23]. Therefore, best labeling strategies employ small dye molecules with direct labeling [93]. In addition to affecting chromatin structure [28,94], stimulation and genetic activity has an influence on the binding kinetics of chromatin [95], a fact that has also been employed for imaging [96].

## 4. Conclusions

With the advent of super-resolution microscopy, researchers have set out to elucidate the highly complex processes involved in regulation of all genetic activities. To a large extent, the role which is played by the three-dimensional arrangement of the chromatin in these regulatory mechanisms today is still unknown. The visualization and quantitative measurement of the distribution of chromatin with high spatial and temporal resolution inside the cell nucleus is, therefore, an essential desideratum in genetics. Since nowadays, super-resolution microscopy methods are available to a large group of researchers, we expect that the combination of HiC-data, computer modelling and super-resolution microscopy data will rapidly further our understanding of the complex processes involved in gene regulation and chromatin organization.

## Figures and Tables

**Figure 2 genes-10-00493-f002:**
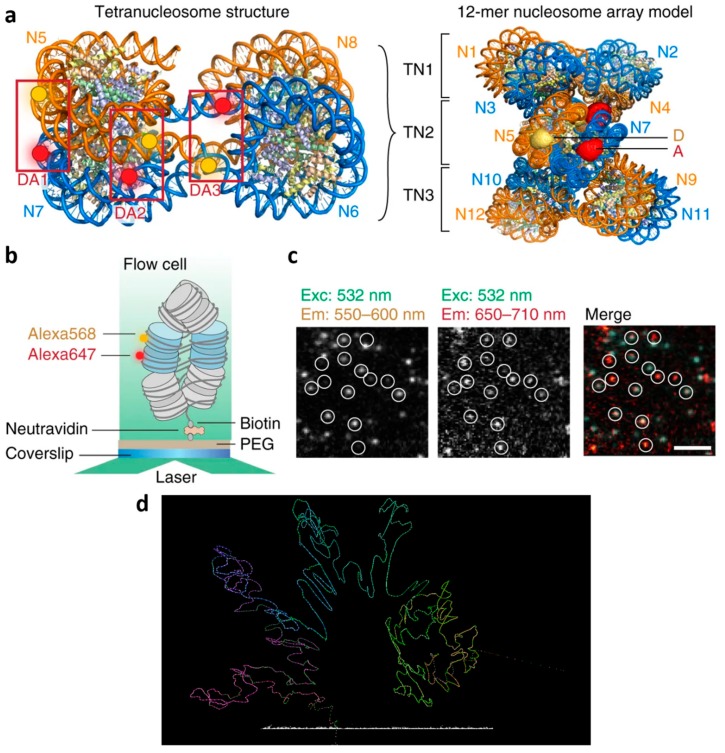
Imaging chromatin conformation. (a–c) Pairs of fluorescent dyes are used to study interactions such as binding between labeled proteins. Single molecule Förster resonance energy transfer (FRET) facilitates the real-time detection of chromatin conformational dynamics. (**a**) Left: A tetranucleosome structure is labeled with three dye pairs DA1, DA2, and DA3. Right: The chromatin fiber can be modeled as a stack of tetranucleosome (TN) units. The middle tetranucleosome carries the fluorescent labels, whose accessible volume is displayed. D donor, A acceptor labels, N nucleosomes. (**b**) Scheme of the total internal reflection fluorescence (TIRF) experiment to measure intra-array single molecule FRET. (**c**) Microscopic images showing FRET data of single chromatin arrays, scale bar: 5 µm. (**d**) Structural data from super-resolution microscopy can be combined with sequencing information (bottom) [91] to yield with higher confidence a model of the underlying structure. (a–c) © 2018 under the Creative Commons license [92] by Kilic et al. [83]. (d) Created with CSynth [91].

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
