# Peer review of "Super-Resolution Microscopy of Chromatin"

_genes, 2019, doi:10.3390/genes10070493_

Round 1

Reviewer 1 Report

Udo Birk's paper on super-resolution microscopy is well written and a good overview of the field suitable for publication in Genes.

My major comment is that the review, while well written and comprehensive, did not seem to clearly provide insight surrounding the specifics of imaging chromatin, vs other applications of super-res microscopy. Could more be made of this, perhaps with the addition of a paragraph towards the end that clearly outlined the specific considerations needed for imaging chromatin, vs other types of SMLM?

It is also somewhat unusual to have a review article on super-resolution without any microscopy images showcasing the resolution improvement and the types of images that can be obtained?

Minor comments:

p1 L11: biological wonder -- perhaps over the top.
p1 L24 - unclear to me here what 'form' is.
p1 L49 as above, I am not exactly sure what form is referring to here.
p2 L66 fool proof, not fool-prove (however nice discussion of caveats here)
p2 l78 - extraordinary -- not necessary, super-res is now quite normal.
In general, the introduction feels a little long, talks a lot about why super-res is useful, could get more quickly to point of showing us what has been done with super-res, specifically as applies to chromatin.
p3 L132 - extraordinary again - feels a little hyperbolic.
p3 L202 - probably should include discussion of DNA-PAINT here, eg Nieves et al, Genes 2018 on applications to SMLM.
p5 L233 - the light sheet community discussion should include a reference, at the very least a GitHub reference for this community software. There are also reviews in general on ML methods and comparative algorithms for superres, eg: Sage et al., Nature Methods 2019 Super-resolution fight club: assessment of 2D and 3D single-molecule localization microscopy software

L327 - there are many more methods for single plane illumination, eg soSPIM, Galland et al. Nature Methods 2015. Perhaps this is what L338 refers to, but no reference?

Author Response

I am grateful for your comments and suggestions, which I am glad to have included in the revised version of the manuscript. The manuscript has surely benefited from these corrections. Thank you!

Reviewer 2 Report

Dr Birk has compiled a very interesting review on recent super-resolution microscopy approaches to unravel the nanoscopic structure of chromatin and to visualise gene regulation. The author presents an introduction to microscopy and its challenges, detailing sample preparation, imaging and image analysis. I found the text very interesting and informative, and, once published, I will distribute it around my group. I was wondering however why there was no illustration present. In my opinion the review would be strengthened with the addition of an illustration highlighting chromatin structure and the challenges that arise when imaging it and how these can be overcome with recently developed methods. Moreover, I was puzzled that ATAC-see wasn't mentioned in the review. I feel that the sections on super-resolution microscopy techniques could be fleshed out with one or two more sentences to also mention light sheet microscopy approaches.

Highlighted below are some minor language issues:

line 33 - coined the term

line 122 - to routinely detect

line 151 - fluorophore is limited

line 154 - induced

line 210 - fluorophores

line 248 - differences

line 252 - have been modified in such a ways

line 343 - isolated

Author Response

(The authors gave the same response as above.)
